Multi-omics data integration reveals the complexity and diversity of host factors associated with influenza virus infection

Zhu Zhaozhong 1 2
You Ruina 1
Li Huiru 1
Feng Shuidong 2
Ma Huan 1
Tuo Chaohao 1
Meng Xiangxian 1
Feng Song fs205@sina.com 3
Peng Yousong pys2013@hnu.edu.cn 1
1 College of Biology, Hunan University , Changsha , China
2 School of Public Health, University of South China , Hengyang , China
3 Xiangya Hospital, Central South University , Changsha , China
Raju Nagarajan
Electronic publication date: 2023 Oct 9
Publication date: 2023
Volume: 11
Electronic Location ID: e16194
Received 2023 Jun 15; Accepted 2023 Sep 6
Copyright: ©2023 Zhu et al.
Copyright year: 2023
Copyright holder: Zhu et al.
License: This is an open access article distributed under the terms of the Creative Commons Attribution License, which permits unrestricted use, distribution, reproduction and adaptation in any medium and for any purpose provided that it is properly attributed. For attribution, the original author(s), title, publication source (PeerJ) and either DOI or URL of the article must be cited.
License URL: https://creativecommons.org/licenses/by/4.0/

Keywords: Influenza virus, Bioinformatics, Multi-omics data

Funding: National Key Plan for Scientific Research and Development of China 2022YFC2303802 National Natural Science Foundation of China 32170651 This study was funded by the National Key Plan for Scientific Research and Development of China (2022YFC2303802) and the National Natural Science Foundation of China (32170651). The funders had no role in study design, data collection and analysis, decision to publish, or preparation of the manuscript.

==============================
Influenza viruses pose a significant and ongoing threat to human health. Many host factors have been identified to be associated with influenza virus infection. However, there is currently a lack of an integrated resource for these host factors. This study integrated human genes and proteins associated with influenza virus infections for 14 subtypes of influenza A viruses, as well as influenza B and C viruses, and built a database named H2Flu to store and organize these genes or proteins. The database includes 28,639 differentially expressed genes (DEGs), 1,850 differentially expressed proteins, and 442 proteins with differential posttranslational modifications after influenza virus infection, as well as 3,040 human proteins that interact with influenza virus proteins and 57 human susceptibility genes. Further analysis showed that the dynamic response of human cells to virus infection, cell type and strain specificity contribute significantly to the diversity of DEGs. Additionally, large heterogeneity was also observed in protein-protein interactions between humans and different types or subtypes of influenza viruses. Overall, the study deepens our understanding of the diversity and complexity of interactions between influenza viruses and humans, and provides a valuable resource for further studies on such interactions.

Introduction

Influenza viruses are segmented RNA viruses that belong to the negative-sense, single-stranded category (Steinhauer & Skehel, 2002; Javanian et al., 2021). They can be classified into four types: A, B, C and D (Krammer et al., 2018). Among them, type A can be further subdivided into various subtypes based on surface proteins hemagglutinin (HA) and neuraminidase (NA), such as H1N1, H3N2, H5N1, H7N9, and others (Krammer et al., 2018; Zhuang et al., 2019). Influenza viruses cause infections in 5–15% of the global population and over 400,000 deaths annually and pose a significant threat to human health (Al Farroukh et al., 2022; Lampejo, 2020). The life cycle of the influenza virus encompasses several essential steps including virus entry, replication of the genomic RNA, translation of mRNA, protein processing, as well as assembly and release of virus particles (Watanabe, Watanabe & Kawaoka, 2010; Peteranderl, Herold & Schmoldt, 2016). A comprehensive understanding of the host cell’s response to influenza virus infection is critical for effective treatment of the viral infections and antiviral drug development  (Lampejo, 2020; De Chassey et al., 2014).

A large number of host factors have been reported to participate in the process of influenza virus infection (Luo, 2012; Moreira, Yamauchi & Matthias, 2021). Although traditional methods have identified several important host factors associated with influenza virus infection such as HLA and TMRPSS2 (McMichael et al., 1977; Cheng et al., 2015), they are generally low-throughput, time-consuming and labor-intensive, which hinder a systematic understanding of the virus-host interactions. Thus, numerous high-throughput experimental and computational methods have been developed to identify host factors associated with influenza virus infections (Friedel & Haas, 2011; Trimarco & Heaton, 2022). Transcriptome and proteome sequencing have been instrumental in identifying numerous host genes and proteins that play crucial roles in virus infections through differential expression analysis (Vijayakumar et al., 2022; Babu & Snyder, 2023). As an illustration, Hancock et al. (2018) identified a total of 1,903 genes differentially expressed in type 2 alveolar epithelial cells infected with H1N1 and revealed that influenza virus downregulated Wnt signaling in the lung. Besides the transcriptome and proteome level, influenza virus infections also disrupt the post-translational modification of host proteins, and proteins with differential post-translation modification may also participate in the virus infection process (Kumar et al., 2020; Söderholm et al., 2016). For instance, the phosphorylation of 1113 proteins was regulated in primary human macrophages after influenza virus infection, highlighting the importance of global phosphoproteomic profiling in primary cells following viral infections  (Söderholm et al., 2016). The high-throughput methods of detecting protein-protein interactions such as yeast two-hybrid experiments and affinity purification-mass spectrometry experiments, are also very helpful in identifying host proteins involved in virus infection (Schaack & Mehle, 2020; Wang et al., 2016). For example, 560 interactions were observed between 79 human proteins and NS1/NS2 proteins of 9 distinct influenza virus strains using the yeast two-hybrid method (Chassey et al., 2013). Whole genome or exome sequencing has also aided in identifying host susceptibility genes by genome-wide association analysis (Tam et al., 2019; Khor & Hibberd, 2012). For instance, the MX1 gene was demonstrated to play a critical role in host’s antiviral defense against influenza H7N9 viruses based on whole-genome sequencing (Chen et al., 2021). Integrating host factors associated with influenza virus infection can facilitate a comprehensive understanding of interactions between influenza viruses and their hosts (De Chassey et al., 2014; Moreira, Yamauchi & Matthias, 2021).

Although databases like H2V contain information about human genes and proteins that respond to infections of multiple viruses such as the Severe Acute Respiratory Syndrome Coronavirus (SARS-CoV) and Middle East Respiratory Syndrome Coronavirus (MERS-CoV) (Zhou, Bao & Ning, 2021), there is currently a lack of an integrated resource for influenza viruses. To address this gap, we undertook the integration of human genes or proteins associated with influenza virus infection at different levels and further developed a dedicated database called H2Flu to effectively organize and store these human factors. This comprehensive effort greatly facilitates research on the role of host factors in virus infections and contributes to the development of antiviral drugs  (De Chassey et al., 2014; Friedel & Haas, 2011).

Material and Methods

Data collection

Five kinds of genes or proteins including differentially expressed genes (DEGs), differentially expressed proteins (DEPs), proteins with differential post-translational modifications (DPMs), proteins that participate in human-virus protein–protein interactions (P-PPIs) and human susceptibility factors (SHFs) associated with infection of human or human cells were obtained for 14 subtypes of influenza A viruses, as well as influenza B and C viruses, from publicly available databases and the literature.

Specifically, DEGs were extracted from the HVIDB (Version 1.0) (Yang et al., 2021) database. The DEPs were manually collected from the Influenza Research Database (IRD) (https://www.fludb.org/) (Zhang et al., 2017) and from literatures in the PubMed database by searching “(influenza[TIAB] OR flu[TIAB]) AND ((proteome[TIAB]) OR (proteomics[TIAB]))” on March 1st, 2023. The DPMs were manually collected from literatures in the PubMed database. The PPIs between influenza viruses and humans were obtained from public databases including Viruses.STRING (Version 10.5)  (Cook et al., 2018), IntAct (Version 4.2.3.2) (Orchard et al., 2014), BioGrid (Version 4.4.208) (Stark et al., 2006), Virhostnet(Version 3.0) (Guirimand, Delmotte & Navratil, 2015), HVIDB (Version 1.0), VirusMINT (Version 1.0) (Chatr-aryamontri et al., 2009), VirusMentha (Version 1.0) (Calderone, Licata & Cesareni, 2015) and HVPPI (Version 1.0) (Li et al., 2022). To ensure the reliability of the data, only experimentally validated PPIs from these databases were selected. The PPIs were organized by virus type or subtypes.

To obtain the SHFs of the influenza virus, we firstly collected 1254 abstracts from the PubMed database (Canese & Weis, 2013) by searching “(influenza[TIAB] OR flu[TIAB]) AND human[TIAB] AND((susceptible[TIAB]) OR (susceptibility[TIAB]) OR (sensibility[TIAB]) OR (sensitiveness[TIAB]) OR (susceptivenes[TIAB]))” on April 13th, 2023. Then, each abstract was manually screened based on whether it contained SHFs, which resulted in 48 abstracts. Finally, the full texts of these abstracts were read carefully and 57 SHFs were compiled from these papers.

Functional enrichment analysis

The Gene Ontology (GO) and Kyoto Encyclopedia of Genes and Genomes (KEGG) pathway enrichment analysis of human genes were conducted with functions of enrichGO() and enrichKEGG() in the package “clusterProfiler” (version 4.8.2) (Wu et al., 2021) and org.Hs.eg.db(version 3.17.0) (Carlson et al., 2019) in R (version 4.0.3) (R Core Team, 2013).

Statistical analysis

All statistical analyses were conducted in R (version 4.0.3) (R Core Team, 2013). The Wilcoxon rank-sum test was conducted using the wilcox.test() function; the linear regression fitting was performed using the lm() function; the correlation coefficient was calculated using the cor.test () function.

Results

Data summary

As shown in Fig. 1A and Table S1, two human influenza A viruses including A(H1N1) and A(H3N2) had a large number of genes or proteins associated with human infection (for clarity, they were defined as Virus-Infection-associated Human Factors (VIHFs)). The former had a total of 12,556 VIHFs which included 10,068 DEGs, 1,435 DEPs, 104 DPMs, 2,654 P-PPIs and 48 SHFs; the latter had 15,857 VIHFs which included 15,250 DEGs, 41 DEPs, 342 DPMs, 534 P-PPIs and seven SHFs. Besides human influenza A viruses, two avian influenza viruses, i.e., A(H5N1) and A(H7N9), also had a large number of VIHFs, with the former having 24,683 VIHFs and the latter having 6,244 VIHFs. The remaining viruses, including H3N8, H4N6, H5N2, H5N3, H5N6, H6N8, H7N1, H7N4, H7N7, H9N2, Influenza B virus and Influenza C virus possessed one, one 286, 83, one, one, one, one 5,093, 128, 21 and six VIHFs, respectively.

Figure 1 Data summary of genes or proteins associated with human infection of influenza viruses. For clarity, these genes or proteins were defined as Virus-Infection-associated Human Factors (VIHFs).

(A) The number of five types of VIHFs in different influenza types or subtypes. (B) The shared ratio of VIHFs between influenza types or subtypes which was colored according to the figure legend on the top left. The shared ratios of VIHFs in the upper- and lower-triangular heatmaps were calculated by taking the total number of VIHFs in the left and top type or subtypes, respectively, as the denominator, and by taking the shared number of VIHFs between the left and top types or subtypes as the nominator. (C) Overlap of VIHFs between DEGs, DEPs, DPMs and P-PPIs. (D) Overlap of VIHFs between SHFs and other kinds of VIHFs. (E) Distribution of data by viral strain, cell type and infection time points. The data above and below the black line referred to the transcriptomic and proteomic data, respectively. The blue stars indicated that data for both transcriptomic and proteomic data were available.

When comparing VIHFs between influenza types or subtypes, as shown in Fig. 1B, we observed a very small proportion of shared VIHFs. Two human influenza A viruses, i.e., A(H1N1) and A(H3N2), had a high ratio of shared VIHFs. Subtypes with the same HA subtype seemed to have a relatively high ratio of shared VIHFs, such as subtypes of H5N1, H5N2 and H5N3, and subtypes of H7N9 and H7N7.

When comparing VIHFs of different types, we found a large difference between them. Although DEGs accounted for most VIHFs, each kind of VIHFs had unique genes. As shown in Fig. 1C, DEPs, DPMs, P-PPIs and SHFs had 180, 257, 508 and five unique genes, respectively. Interestingly, there were some genes shared among multiple kinds of VIHFs. For example, 19 genes were shared between DEGs, DEPs, DPMs and P-PPIs, suggesting that they may play important roles in influenza virus infections. As shown in Fig. 1D, over 90% of SHFs overlapped with other kinds of VIHFs, indicating that most susceptible genes were disrupted after influenza virus infection.

The meta information of VIHFs including the relavant viral strains, cell type and infection time points were also provided (Fig. 1E). Transcriptomic data from 14 influenza virus strains infecting 8 cells at 16 different time points, and proteomic data from seven influenza virus strains infecting 3 cell lines at 8 time points, were compiled separately.

Dynamic response of human cells to influenza virus infection

The dynamic response of humans to influenza virus infection was analyzed since there were a large number of DEGs. The number of DEGs identified at different time points post infection was analyzed for 14 viral strains which infected 8 different cell types (Fig. 2A). It was found that there was a small number of DEGs within 10 h post infection(hpi); then, the number of DEGs increased rapidly from 10 to 20 hpi, reaching a peak around 20 hpi; then, it began to decrease and kept stable after 30 hpi. When comparing the shared DEGs between different time points, we found that there was a negative correlation (Pearson Correlation Coefficient = −0.30) between the shared ratio of DEGs and the size of time intervals (Fig. 2B). DEGs in different time points had a small ratio of overlaps except those after 24 hpi (Fig. 2C), indicating distinct DEGs at different stages of virus infections.

Figure 2 Dynamic response of human cells to influenza virus infection.

(A) The number of DEGs at different time points after infection. (B) The correlation analysis between the shared ratio of DEGs and the size of time intervals. The black line referred to the linear regression fitting and the gray area referred to the 95% confidence interval of the regressed line. (C) The shared ratio of DEGs among different time points after influenza virus infection which was colored according to the figure legend on the top right. The shared ratios of DEGs in the upper and lower triangular heatmaps were calculated by using the total number of DEGs in the left and top time points as the respective denominators, and by taking the shared number of DEGs between the left and top time points as the nominator.

The cell specificity contributed larger to the diversity of DEGs than the strain specificity

We further analyzed the strain and cell type specificity for DEGs. As shown in Fig. 3A, different strains of influenza viruses had varying numbers of DEGs with the median number of DEGs ranging from 22 to 4,798. Even within the same subtype such as A(H1N1), DEGs in different strains also differed much. Interestingly, two strains of highly pathogenic avian influenza H5N1 viruses which led to human infection and death, i.e., A/Vietnam/1203/2004 and A/Vietnam/UT3028II/03, had the largest number of DEGs, while avian influenza viruses such as A/duck/Malaysia/F118/08/2004 and A/duck/Malaysia/F119/3/1997 generally had a small number of DEGs. To remove the influence of cell type, we also compared the number of DEGs between virus strains which infected the same cell type. As shown in Fig. S1, large differences were still observed between the number of DEGs of viral strains infecting the same cell type, even for the strains of the same subtype such as four strains of A(H1N1) infecting the A549 cell.

Figure 3 Analysis of the strain and cell specificity in DEGs.

(A) Number of DEGs in different influenza virus strains. (B) Number of DEGs in different cell types infected by influenza viruses. (C) The shared ratio of DEGs between different strains and cell types. Asterisks (**) indicate p-value < 0.01.

Similarly, we observed large differences between DEGs in different human cell types after influenza virus infections (Fig. 3B). The median number of DEGs ranged from 55 to 6,710 across eight human cell types after influenza virus infections. Overall, when comparing the shared ratios of DEGs between different strains and different cell types, we found that the latter was smaller than the former, suggesting the cell specificity was greater than strain specificity in DEGs, and both of them contributed significantly to the diversity of DEGs in influenza virus infections.

The heterogeneity of P-PPIs in influenza viruses

The P-PPIs were the second largest type of VIHFs. Thus, the heterogeneity of P-PPIs in different types or subtypes of influenza viruses was analyzed. As shown in Fig. 4A, the shared ratios of P-PPIs between influenza virus types or subtypes were generally smaller than 0.1, suggesting different influenza viruses had large differences between P-PPIs. Further analysis of the P-PPIs by proteins of the influenza virus still showed large differences between different viruses (Fig. 4B).

Figure 4 The heterogeneity of P-PPIs in different influenza virus types or subtypes.

(A) The shared ratio of P-PPIs between different influenza virus types or subtypes which was colored according to the figure legend on the top left. The shared ratios of P-PPIs in the upper- and lower-triangular heatmaps were calculated by taking the total number of P-PPIs in the left and top type or subtypes, respectively, as the denominator, and by taking the shared number of P-PPIs between the left and top types or subtypes as the nominator. (B) The shared ratio of P-PPIs between influenza viruses by influenza virus proteins. Only the P-PPIs that interact with the virus protein were considered when calculating the shared ratio of P-PPIs between a pair of influenza virus types or subtypes.

Overview of the H2Flu

A database named H2Flu was created to store and organize different types of VIHFs of influenza viruses. The database is publicly available at http://computationalbiology.cn/H2Flu. It mainly includes pages of Homepage, Browse, Search, Statistic, Download and Contact us(Fig. 5).

Figure 5 The structure of H2Flu database.

Homepage.

This page included simple introductions of the database, rapid links to the Search page, news of the database and useful links.

Browse.

The page displayed VIHFs by influenza virus types or subtypes, or by data types. A given type of VIHFs for a given virus type or subtype would be shown in a table, based on which gene function enrichment analysis would be conducted.

Search.

Users can search for a VIHF by gene name, which would output the structure, tissue specificity, related diseases and candidate drugs targeting the VIHF, or users can search a combination of a given data type and a given virus type or subtype, which would output a list of VIHFs that can be further used in gene function enrichment analysis.

Statistic.

This page displayed a summary of statistics about the VIHFs by virus type or subtype, or by data type.

Download.

All VIHFs used in the database can be easily downloaded on the page.

Contact us.

To ensure quick communications with the authors in case users encounter any issues, the email address of the H2Flu database developers and the laboratory’s addresses were provided.

An application case of H2Flu

We illustrated the potential usage of H2Flu in integration of multi-omic data in influenza virus infection by comparing the DEGs and DEPs during the infection of Calu-3 cells at 18 hpi by a viral strain of H5N1 virus (A/Vietnam/1203/2004). As shown in Fig. 6A, there were only two genes that were both up-regulated at the transcription and proteome level, suggesting the large difference of upregulated genes in host response to viral infection at different levels. However, more than one-third of downregulated DEPs were also found to be downregulated at the transcriptional level, suggesting the consistent role of these genes in the viral infection. Functional analysis of these genes showed that they were enriched in biological processes related to protein folding, localization, regulation of protein stability, and in KEGG pathways related to metabolisms of carbon and amino acid, actin cytoskeleton, focal adhesion, and so on.

Figure 6 Integration analysis of DEPs and DEGs in Calu-3 cells at 18 hpi after infection with the A/Vietnam/1203/2004 strain.

(A) Overlap between upregulated DEPs and DEGs in Calu-3 cells after infection with the A/Vietnam/1203/2004 strain. (B) Same as (A) for downregulated DEPs and DEGs. (C) GO-term functional enrichment by three categories (BP, MF, CC) and KEGG pathway analysis were performed for the concurrently downregulated DEGs and DEPs.

Discussion

As research on the influenza virus progresses, there is a growing emphasis not only on the virus itself, but also on the human factors involved in virus infections (Sladkova & Kostolansky, 2006; Ali et al., 2022). However, the majority of research focuses on some kinds of host factors such as DEGs for only one or a few influenza viruses (Hancock et al., 2018; Schaack & Mehle, 2020). To bridge this gap, we have successfully integrated human genes and proteins at different levels that are associated with infections of more than ten types or subtypes of influenza viruses and further built a database known as H2Flu. This database serves as an organized and easily accessible resource for the scientific community, providing valuable insights for further research on the complex relationship between influenza viruses and humans.

The most important discovery of the study is the large heterogeneity of human factors associated with influenza virus infections. The large difference between different types of VIHFs (Figs. 1C & 1D) suggested that the influenza virus may disrupt the host system at multiple levels from transcriptome, proteome and post-translation modification to interactome. Thus, it is necessary to integrate multi-omics methods for a systematic understanding of the virus-host interactions (Babu & Snyder, 2023; Tang et al., 2022). The dynamic changes in DEGs following influenza virus infection suggest different genes or proteins involved in different stages of viral infection, which is consistent with previous studies (Sladkova & Kostolansky, 2006; Ali et al., 2022). Interestingly, both strain specificity and cell specificity contributed much to the heterogeneity of DEGs, which suggests that it is necessary to consider both the viral strain and cell type when comparing and integrating DEGs in different experiments.

The largest limitation of the study is the limited and biased data in the database. Firstly, only 16 types or subtypes of influenza viruses were used in the database, and most VIHFs were identified in some types such as A(H1N1) and A(H3N2). There were more than 100 subtypes of influenza A viruses (Krammer et al., 2018; Zhuang et al., 2019). Thus, VIHFs of other subtypes of influenza A viruses should be added to the database in the future. Secondly, only proteins or protein-coding genes were collected in the database. Actually, RNAs also take part in virus infections  (Liao et al., 2022; Guo et al., 2022). Thus, RNAs associated with influenza virus infections should be also added to the database in the future. Thirdly, only five kinds of VIHFs were included in the database and there was a serious imbalance of these data types. More VIHFs at other levels such as the epigenetic level can be added to the database as the influenza virus has been reported to influence the epigenetic states of host cells. Nevertheless, this study systematically compiled human genes and proteins associated with influenza virus infection in human cells, and further built a database named H2Flu to store these genes. It deepens our understanding of the diversity and complexity of interactions between influenza viruses and humans, and provides a valuable resource for further studying such interactions.

Supplemental Information

Supplemental Information 1 Supplementary Materials

Click here for additional data file.

We thank the members of PengLab for helpful discussions on the manuscript.

Additional Information and Declarations

Competing Interests

Author Contributions

Data Availability

The authors declare there are no competing interests.

Zhaozhong Zhu conceived and designed the experiments, performed the experiments, analyzed the data, prepared figures and/or tables, authored or reviewed drafts of the article, and approved the final draft.

Ruina You performed the experiments, analyzed the data, authored or reviewed drafts of the article, and approved the final draft.

Huiru Li performed the experiments, analyzed the data, authored or reviewed drafts of the article, and approved the final draft.

Shuidong Feng performed the experiments, analyzed the data, authored or reviewed drafts of the article, and approved the final draft.

Huan Ma performed the experiments, analyzed the data, authored or reviewed drafts of the article, and approved the final draft.

Chaohao Tuo performed the experiments, analyzed the data, authored or reviewed drafts of the article, and approved the final draft.

Xiangxian Meng performed the experiments, analyzed the data, authored or reviewed drafts of the article, and approved the final draft.

Song Feng performed the experiments, analyzed the data, authored or reviewed drafts of the article, and approved the final draft.

Yousong Peng conceived and designed the experiments, performed the experiments, prepared figures and/or tables, authored or reviewed drafts of the article, and approved the final draft.

The following information was supplied regarding data availability:

The H2Flu database is available at: http://computationalbiology.cn/H2Flu.

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
