# Peer review of "Multi-omics data integration reveals the complexity and diversity of host factors associated with influenza virus infection"

_PeerJ, doi:10.7717/peerj.16194_

## Round 0.1 · original submission · Minor Revisions

Please go through the comments from the reviewers and make needed changes in the revised version of the manuscript.

·

Basic reporting

Check additional comments

Experimental design

Check additional comments

Validity of the findings

Check additional comments

Additional comments

In this work, the authors aim to build a database named H2Flu to include influenza virus infection related genes and proteins. The data were collected from various existing databases and literature sources. The reviewer acknowledges the potential value of this database for future influenza virus research. However, there are several concerns and suggestions that the authors should address.

Major issue:
1) The authors collected different types of omics data in the database. But all the analyses provided by authors focused on single-omics analysis, the manuscript could benefit from including one application of the database on multi-omics data analysis.
2) Some important information is missing in the manuscript, for example, in the section “Dynamic response of human cells to influenza virus infection”, where do the time data, virus strains and cell types information come from? (Line 133-134) This information should be included in the “data summary” section, I suggest authors to include a figure in this section to show the variables included in the database.
3) The links in the manuscript do not work.

Minor issue:
1) Line 43, “the low efficiency of these methods …”, could authors elaborate a little bit on this to explain the drawbacks of the traditional methods.
2) Line 80, “14 subtypes of influenza A virus, as well as influenza B and C viruses, …”, the authors mentioned there is also type D for influenza viruses, why not include type D?
3) Line 83, “… manually collected from literatures in the PubMed database and IDB database …”, please include the criterion for selecting the literature like what you have done for PPIs, and what’s the full name for IDB? Also, please include a reference here for the IDB database.
4) Line 88, “only experimentally validated PPIs from the Virus.STRING database were selected”, the authors mentioned several different databases for collecting the PPIs information, only experimentally validated PPIs from the Virus.STRING database was selected, what about other databases?
5) Line 103-108, repetitive information.
6) Line 119-123, please include Figure 1B here if you describe the results of 1B.
7) Line 124-130, same here for Figures 1C and 1D.
8) Figure 1, “… by taking the total number of VIHFs in the left and top types or subtypes, respectively, as the denominator.” It’s quite confusing here, what’s the nominator? Please make it clear. Same for Figures 2 and 4.
9) Figure 4, “The shared ratio of P-PPIs between influenza viruses by influenza virus proteins.” I am confused, what does each column mean? Please make it clear.

Reviewer 2 ·

Basic reporting

The manuscript is clearly written. Adequate references are included and a satisfactory background section is provided. The figures and tables are professional.

Experimental design

The research question is interesting and meaningful. The investigation is rigorous. However, I believe readers could benefit from more detail regarding the cell type samples included. It is likely that different cell types respond differently to influenza infection. Therefore, for the results to be valid, comparisons of results between different strains must be done using the same human cell types. I believe this is how the authors have done (Figure 3) but it can be clarified further for the reader.

Validity of the findings

Preferably, the authors would share the underlying raw data with readers.

Additional comments

Thank you for inviting me to review this manuscript. I believe the manuscript would benefit from some minor clarifications but is otherwise well-written.

Reviewer 3 ·

Basic reporting

The article was written clearly with professional English. Literature and references as well as the figures are appropriate. The overall approach on the interactome of inflenza viruses is interesting and useful. Language needs some polishing at different points (Authors use "for example" too often)

Experimental design

The database presented in this article is relevant and meaningful, it provides an integrative approach on the interactome of influenza viruses with potential implications in potential anti-viral discoveries. Methods were explained sufficiently. The database presented here fills an important gap in the field.

Validity of the findings

Results and analyses provided in this article are meaningful. They also sound statistically empowered. Conclusions and discussions is appropriate.

Reviewer 4 ·

Basic reporting

The use of clear, unambiguous, and professional English language is consistently maintained throughout the study. The introduction and background sections provide necessary context, and the literature is thoroughly referenced and relevant to the topic. The structure of the study adheres to the standards set by PeerJ and aligns with the norms of the respective discipline. The figures presented in the study are highly relevant, of high quality, and effectively labeled and described. Additionally, raw data has been provided.
However, the line 67, "Severe acute respiratory syndrome coronavirus and Middle East respiratory syndrome coronavirus" need revision using the correct capitalization for viruses if come as proper nouns.

Experimental design

Original primary research presented which falls within the scope of the journal. The research question is clearly defined, relevant, and meaningful. The study explicitly addresses how it fills a recognized gap in knowledge. A thorough investigation is conducted to adhere to rigorous technical and ethical standards. The methods employed are adequately described, providing enough detail and information to enable replication.

Validity of the findings

Assessment of impact and novelty was conducted. The rationale and benefit to the existing literature are clearly explained. All the underlying data have been provided and they are reliable, statistically sound, and controlled. The conclusions are clearly expressed, connected to the original research question, and limited to supporting the results. Limitations have been discussed well by the authors and satisfactory suggestions for further research were included.

Additional comments

This study systematically compiled human genes and proteins that are associated with influenza virus infection in human cells. Additionally, a database named H2Flu was constructed to store these genes.
Effective in silico experiments were conducted using relevant literature. Since, the research presents novel approach to fill identified gap in in exploring the human genes and proteins that are associated with influenza virus infection in human cells. in the So, there is no hesitation from my side to recommend this article for publication.

Annotated reviews are not available for download in order to protect the identity of reviewers who chose to remain anonymous.

---

## Round 0.2 · accepted · Accept

The manuscript is acceptable.

·

Basic reporting

no comment

Experimental design

no comment

Validity of the findings

no comment

Additional comments

no comment